# The *Fabp4*-Cre-Model is Insufficient to Study *Hoxc9* Function in Adipose Tissue

**DOI:** 10.3390/biomedicines8070184

**Published:** 2020-06-29

**Authors:** Sebastian Dommel, Claudia Berger, Anne Kunath, Matthias Kern, Martin Gericke, Peter Kovacs, Esther Guiu-Jurado, Nora Klöting, Matthias Blüher

**Affiliations:** 1Medical Department III—Endocrinology, Nephrology, Rheumatology, University of Leipzig Medical Center, D-04103 Leipzig, Germany; claudia.berger@medizin.uni-leipzig.de (C.B.); Kunath.anne@gmx.de (A.K.); matthias.kern@medizin.uni-leipzig.de (M.K.); peter.kovacs@medizin.uni-leipzig.de (P.K.); esther.guiujurado@medizin.uni-leipzig.de (E.G.-J.); nora.kloeting@medizin.uni-leipzig.de (N.K.); 2Institute of Anatomy, Leipzig University, D-04103 Leipzig, Germany; martin.gericke@medizin.uni-leipzig.de; 3Institute of Anatomy and Cell Biology, Martin-Luther-University, D-06108 Halle (Saale), Germany; 4Helmholtz Institute for Metabolic, Obesity and Vascular Research (HI-MAG) of the Helmholtz Zentrum München at the University of Leipzig and University Hospital Leipzig, Leipzig University, D-04103 Leipzig, Germany

**Keywords:** developmental genes, adipose tissue, obesity, adipocytes

## Abstract

Developmental genes are important regulators of fat distribution and adipose tissue (AT) function. In humans, the expression of homeobox c9 (*HOXC9*) is significantly higher in subcutaneous compared to omental AT and correlates with body fat mass. To gain more mechanistic insights into the role of *Hoxc9* in AT, we generated *Fabp4*-Cre-mediated *Hoxc9* knockout mice (AT*Hoxc9*^-/-^). Male and female AT*Hoxc9*^-/-^ mice were studied together with littermate controls both under chow diet (CD) and high-fat diet (HFD) conditions. Under HFD, only male AT*Hoxc9*^-/-^ mice gained less body weight and exhibited improved glucose tolerance. In both male and female mice, body weight, as well as the parameters of glucose metabolism and AT function were not significantly different between AT*Hoxc9*^-/-^ and littermate control CD fed mice. We found that crossing *Hoxc9* floxed mice with *Fabp4*-Cre mice did not produce a biologically relevant ablation of *Hoxc9* in AT. However, we hypothesized that even subtle reductions of the generally low AT *Hoxc9* expression may cause the leaner and metabolically healthier phenotype of male HFD-challenged AT*Hoxc9*^-/-^ mice. Different models of in vitro adipogenesis revealed that *Hoxc9* expression precedes the expression of *Fabp4*, suggesting that ablation of *Hoxc9* expression in AT needs to be achieved by targeting earlier stages of AT development.

## 1. Introduction

An important breakthrough in developmental biology was the discovery that there are common genes that control early embryonic development in organisms ranging from the fruit fly to humans. Edward B. Lewis demonstrated that homeotic genes are of central importance for segmentation in *Drosophila melanogaster* [1]. Homeotic genes are lined up on the DNA in exactly the same order as they are expressed along the body axis during embryogenesis. As a subset of homeotic genes, Hox genes are considered as master regulators for segmentation patterning during embryogenesis. Humans and mice possess thirty-nine Hox genes organized into four clusters (A to D) on four chromosomes (*Hoxa* 7p15, *Hoxb* 7q21.2, *Hoxc* 12q13, and *Hoxd* 2q31). Each cluster consists of thirteen paralog groups with nine to eleven members [2]. On each chromosome, the Hox clusters are organized in 3′ to 5′ orientation with paralogous lower number group members (e.g., *Hoxa1*, *Hoxb1*, *Hoxd1*) at the 3′ end, whereas higher number members are located more in the 5′ direction. During embryogenesis, Hox genes are expressed in a spatial-temporal manner. According to their chromosomal localization, 3′ genes are expressed in more anterior body regions, while in posterior areas, 5′ Hox genes are paramount. Besides, anterior members are expressed earlier than posterior [3,4]. In addition, they act as transcriptional regulators in adult organisms. For this, all homeotic genes are united by the presence of a highly conserved 180 bp DNA-binding domain, called homeobox [5].

Previously, we analyzed expression patterns of developmental control genes in murine adipocytes and stroma vascular fraction (SVF) from both subcutaneous (scWAT) and epididymal white AT (eWAT) depots. Thereby, we detected twelve genes being differentially expressed between both depots, with seven of them showing higher expression levels in eWAT and five genes more expressed in scWAT [6]. Among those AT-expressed developmental genes (*Tbx15*, *Shox2*, *En1*, *Sfrp2*), we found significantly higher *HOXC9* expression in human subcutaneous compared to visceral AT correlating with body fat mass and adipocyte size [6,7]. However, the precise mechanisms by which *Hoxc9* may contribute to obesity, AT distribution, and function are not well understood. Because germ line mutations in Hox genes are lethal for the developing embryo, we used the adipocyte-specific Cre-recombinase under control of the fatty acid-binding protein 4 (*Fabp4*) promotor to create mice lacking *Hoxc9* in AT (AT*Hoxc9*^−/−^) [8,9]. Subsequently, we characterized the consequences of *Hoxc9* deletion in AT on the morphological and metabolic parameters of AT*Hoxc9*^-/-^ mice up to an age of 30 weeks.

## 2. Experimental Section

### 2.1. Animal Care and Research Diets

All animal studies were approved by the local authorities of the state of Saxony, Germany, as recommended by the responsible local animal ethics review board (Landesdirektion Leipzig, TVV15/16, T07/16, T02/19). All mice were housed in pathogen-free facilities in groups of three to five at 22 ± 2 °C on a 12 h light/dark cycle. Animals were bred in laboratories at Leipzig University and were fed a standard chow diet (CD, Ssniff Spezialdiäten, Soest, Germany) containing 38.2% carbohydrates, 21.2% proteins, and 3.8% fat (10% calories from fat). Diet-induced obesity was achieved by feeding a high-fat, high-sucrose diet (HFD, Diet Identification No. D12492; Research Diets, produced by Ssniff Spezialdiäten) containing 26.2% carbohydrates, 26.3% protein, and 34.9% fat (60% calories from fat) starting at 6 weeks of age. All animals had *ad libitum* access to food and water at all times, except for experiments where a fasting state was required. 

### 2.2. Generation of AT-Specific Hoxc9 Knockout Mice

Floxed *Hoxc9* mice (*Hoxc9^lox/lox^*) were generated by TaconicArtemis (TaconicArtemis GmbH, Cologne, Germany) (Figure 1). The loxP-flanked *Hoxc9* allele was generated in C57BL/6NTac embryonic stem cells by transfecting them with the targeting vector. Besides floxed *Hoxc9* exons 2 and 3, the vector contained thymidine kinase as negative and two positive selection markers: an FRT flanked neo cassette and a puro cassette circumscribed by F3 sites. After removal of positive selection markers in vivo by Flp-mediated recombination, AT*Hoxc9*^-/-^ were derived by crossing *Hoxc9*^lox/wt^ mice with *Hoxc9*^lox/wt^ mice expressing the Cre recombinase under the control of the *Fabp4* promoter/enhancer [10]. In AT, Cre recombinase mediates the deletion of all loxP-flanked alleles, resulting in an AT-specific *Hoxc9* knockout (AT*Hoxc9*^-/-^). Animals were backcrossed on a C57BL/6NTac background for at least five generations, and both sexes were included in all experiments. 

### 2.3. Molecular Characterization and Genotyping of ATHoxc9^-/-^ Mice

Genotyping was performed by PCR using genomic DNA isolated from the tail tip or ear stamp by using the DirectPCR (Tail) Lysis Reagent (Viagen Biotech Inc., Los Angeles, CA, USA). Primer pairs for genotyping are listed in Appendix A. PCR was performed as follows: initial denaturation at 95 °C for 15 min, following 35 cycles of denaturation 95 °C for 30 s, annealing 60 °C for 30 s, elongation 72 °C for 1 min, and final elongation 72 °C for 10 min using the FastStart PCR Master (Roche, Basel, Switzerland), primer (biomers.net, Ulm, Germany) and a Peltier Thermal Cycler PTC-200 (Bio-Rad, Hercules, CA, USA). On 2% agarose gel, wild-type mice showed a 205 bp band, whereas *Hoxc9*^lox/lox^ mice produced a 324 bp band. Additionally, AT*Hoxc9*^-/-^ mice showed a 100 bp band for the presence of Cre recombinase.

DNA from AT samples was isolated using the NucleoSpin DNA Lipid Tissue Kit (Macherey-Nagel GmbH & Co. KG, Düren, Germany) according to the manufacturer’s instructions. DNA concentration was quantified with a NanoVue Plus spectrophotometer (GE Healthcare, Chicago, IL). For PCR reactions, 75 ng of DNA were used in combination with primer pairs for *Hoxc9* loxP sites, Cre recombinase, and *Hoxc9* intron 1 exon 2 junction (Appendix A). The following PCR conditions were used: initial denaturation 95 °C for 3 min followed by 30 cycles of denaturation 95 °C for 30 s, annealing 60 °C for 30 s, elongation 72 °C for 30 s, and final elongation 72 °C for 10 min. 

### 2.4. Phenotypic Characterization

All experimental procedures were conducted both in male and female mice. In this study, 19 male and 20 female *ATHoxc9*^-/-^ mice were obtained and compared to 11 male and 16 female control littermates (*Hoxc9*^lox/lox^). Mice were studied from an age of 4 weeks up to 30 weeks under CD or up to 26 weeks of age on HFD. Body weight was recorded weekly, and whole-body fat and lean mass were determined with the EchoMRI700™ instrument (Echo Medical Systems, Houston, TX, USA) at the respective study’s end. At an age of 16 weeks, both sexes underwent a food intake measurement over a time period of 1 week. The daily food intake was calculated as the average intake of chow within the time stated. Intraperitoneal insulin tolerance tests (i.p. ITTs) and glucose tolerance tests (i.p. GTTs) were performed at the age of 25 weeks as previously described [11]. Hyperinsulinemic-euglycemic clamp was performed at 23 - 25 weeks of age as described before [11]. In subgroups, whole body energy metabolism was investigated using an indirect metabolic chamber system (CaloSys V2.1, TSE Systems, Bad Homburg, Germany) at an age of 25 weeks (HFD) or 29 weeks (CD). In brief, 6 to 8 *ATHoxc9*^-/-^ and *ATHoxc9*^lox/lox^ mice of each sex and diet (CD and HFD) were housed for 72 h in metabolic chambers as previously described [11]. Body length (naso-anal length) and rectal body temperature (TH-5 Thermalert Monitoring Thermometer Physitemp, Clifton, NJ, USA) were measured at the end of observation period. Mice were sacrificed at the age of 30 weeks (CD) or 26 weeks (HFD) by an overdose of anesthetic (Isoflurane, Baxter, Unterschleißheim, Germany). Liver, brown (BAT), inguinal (ingWAT), and epigonadal white adipose tissue (eWAT) were removed, weighed, and immediately stored in liquid nitrogen or in 4% formalin for histological investigations. Relative organ weights (liver, BAT, ingWAT, and eWAT) were calculated in relation to body weight.

### 2.5. Blood and Serum Analytical Procedures

Fasting blood glucose levels were obtained from whole venous blood samples using FreeStyle Freedom lite (Abbott GmbH, Ludwigshafen, Germany). 20 µL of whole blood were collected in EDTA-containing tubes for HbA1c analyses (COBAS 7000, Roche, Basel, Switzerland). Insulin (Mouse Ultrasensitive Insulin ELISA, ALPCO, Salem, NH), leptin (Mouse Leptin ELISA Kit, CRYSTAL CHEM INC., Downers Grove, IL, USA), and adiponectin (Adiponectin (mouse) ELISA Kit, AdipoGen^®^ LIFE SCIENCES, Liestal, Switzerland) serum concentrations were measured by ELISA using mouse serum according to the manufacturer’s guidelines. Serum protein levels were determined by OLINK proteomics (Uppsala, Sweden).

### 2.6. Histology

AT was fixed in 4% buffered formaldehyde, rinsed with water, and dehydrated in a graded series of 70–100% ethanol followed by ROTI^®^Histol (Carl Roth GmbH, Karlsruhe, Germany) and paraffin. Multiple 5 µm sections (separated by 80 μm after 3 sections) were obtained from ingWAT and eWAT pads, H&E stained, and analyzed systematically with respect to adipocyte size using a Keyence BZ-X800 microscope and BZ-X800 Analyzer software (Keyence Corp., Osaka, Japan). At least 750 (CD) or 1000 (HFD) adipocytes were analyzed for each genotype to determine the cell size distribution.

### 2.7. RNA Isolation and Tissue-Specific mRNA Expression

Frozen AT was lysed using QIAzol Lysis Reagent (Qiagen GmbH, Hilden, Germany) and Precellys Homogenizer (Bertin Technologies, Montigny-le-Bretonneux, France) for 2 × 20 s, 5000 rpm. RNA was isolated from tissue homogenates with RNeasy Lipid Tissue mini Kit (Qiagen GmbH, Hilden, Germany). Quantity and quality were determined using a NanoVue Plus spectrophotometer (GE Healthcare, Chicago, IL, USA). 2 µg of total RNA were reverse transcribed using Random Primers (Invitrogen, Carlsbad, CA, USA) and Superscript II Reverse Transcriptase (Invitrogen, Carlsbad, CA, USA). Quantitative real-time PCR (qRT-PCR) was performed applying a LightCycler 480 system and LightCycler 480 SYBR Green I Master (Roche, Basel, Switzerland). *Hoxc9* mRNA expression was calculated relative to *36b4* RNA using the Pfaffl method [12]. Primers are listed in Appendix A.

### 2.8. Western Blot Analysis

For Western blot analyses, frozen tissues were homogenized in the radioimmunoprecipitation assay buffer containing a complete ULTRA protease inhibitor cocktail tablet (Roche, Basel, Switzerland) per 10 mL with Precellys Homogenizer (Bertin Technologies, Montigny-le-Bretonneux, France). The homogenate was centrifuged for 15 min at 4 °C and 10,000 rpm. Protein concentration was measured using ROTI^®^Quant (Carl Roth GmbH, Karlsruhe, Germany) and a Tecan Sunrise microplate reader (Tecan Group Ltd., Männedorf, Switzerland). 20 µg of each sample were separated by SDS-PAGE using 4 - 20% Mini-PROTEAN TGX Precast Protein Gels (Bio-Rad, Hercules, CA, USA) and transferred to Amersham^TM^ Hybond^TM^ PVDF membranes (GE Healthcare, Chicago, IL, USA). Non-specific protein binding was blocked with 5% (*w*/*v*) BSA in TBS-T for 1 h. The following antibodies were incubated at 4 °C overnight to detect specific protein expression: β-actin (Sigma, A5060, 1:500), Fabp4 (Abcam, ab92501, 1:2000), Hoxc9 (ThermoFisher Scientific, PA-67618, 1:500), and vinculin (Abcam, ab129002, 1:10,000), followed by an incubation of 1 h at RT with goat anti-rabbit IgG HRP-conjugated (Cell Signaling, CS7074, 1:2000). Protein-antibody interactions were visualized using Pierce ECL Western Blotting Substrate (Thermo Fisher, Waltham, MA, USA) and a g:box system (Syngene, Cambridge, UK).

### 2.9. Post-Natal Investigations of Hoxc9 and Fabp4 mRNA and Protein Expressions

IngWAT was dissected from wild-type C57BL/6NTac mice of 0, 2, 5, 10, 15, and 20 days of age, immediately frozen in liquid nitrogen, and stored at −80 °C. RNA and proteins were isolated, and levels of gene and protein expression were measured as described under Section 2.7 and Section 2.8.

### 2.10. Flow Cytometry Analysis

Flow cytometry analyses were performed as described in Braune et al. [13]. In brief, freshly dissected eWAT from chow-fed animals was digested using collagenase type II (Worthington Biochemical, Lakewood, NJ, USA). The resulting cell suspension was filtered through a 70 µm mesh, followed by an erythrocyte lysis using BD lysis buffer (BD Biosciences, Franklin Lakes, NJ, USA), and afterwards, a blocking of Fc receptors by anti-CD16/32 (1:100, Invitrogen, Carlsbad, CA, USA) for 10 min on ice was performed. Leukocytes were stained by anti-CD45-APC-eFluor 780 (1:100, Invitrogen, Carlsbad, CA, USA). Macrophage populations were stained by anti-F4/80-PE-Cy7 (1:100, Invitrogen, Carlsbad, CA, USA), as well as anti-CD11c-PE (1:100, eBioscience/Thermo Fisher, Waltham, MA, USA) and anti-CD206-Alexa Fluor 647 (1:50, Bio-Rad, Hercules, CA, USA). Lymphocyte populations were stained by anti-CD3-PE-Cy7 (1:50, BioLegend, San Diego, CA, USA), anti-CD4-PE (1:100, BioLegend, San Diego, CA, USA), and anti-CD8b-Alexa Fluor 647 (1:100, BioLegend, San Diego, CA, USA). Isotype controls were implemented by using Armenian hamster IgG isotype control-PE (1:100, eBioscience/Thermo Fisher, Waltham, MA, USA), rat IgG2a negative control-Alexa Fluor 647 (1:50, AbD serotec/Bio-Rad, Hercules, CA, USA), IgG2a, κ isotype control-PE (1:100, BioLegend, San Diego, CA, USA), or IgG2b, κ isotype control-Alexa Fluor 647 (1:100, BioLegend, San Diego, CA, USA). All antibody incubations were done in the dark for 20 min on ice. Finally, DNA staining was realized by 7-amino-actinomycin D (7-AAD). Viable CD45^+^ and F4/80^+^ cells were defined as adipose tissue macrophages (ATMs). Subpopulations could be further distinguished into M1 (CD11c^+^; CD206^−^) and M2 (CD11c^−^; CD206^+^). Viable CD45^+^ and CD3^+^ cells were defined as T lymphocytes, further divided into T helper cells (T_H_; CD4^+^; CD8^−^) and cytotoxic T cells (T_C_; CD4^−^; CD8^+^). Analysis was performed using LSR II (BD Biosciences, Franklin Lakes, NJ, USA) and FACSDiva software 8.0. Quantification was performed using FlowJo 10.0.5 (Tree Star, Ashland, OR, USA).

### 2.11. Cell Culture

For in vitro experiments, 3T3-L1 cells, immortalized epididymal and inguinal adipocytes (kindly provided by Prof. Johannes Klein, Lübeck, Germany), as well as primary cells of the stromal vascular fraction (SVF) were used. SVF cells were gained from scAT of wild-type C57BL/6NTac mice. AT was dissected, lymph nodes removed, and the tissue transferred into DMEM containing gentleMACS^TM^ tubes, minced and digested by collagenase using a gentleMACS^TM^ Octo Dissociator with Heaters (Miltenyi Biotec GmbH, Teterow, Germany) at 37 °C for 40 min. The resulting homogenate was spun down for 10 min at 300 rpm, and sedimented SVF cells were cultured like 3T3-L1 cells. 3T3-L1 cells were cultured and differentiated as described earlier [14]. Immortalized epididymal and inguinal adipocytes were cultured according to the protocols reported by Klein et al. [15,16]. Differentiation was initiated after cell layers reached 100% confluence. Cell lines were harvested at 80% pre-confluence, at day 0 (= day of induction, 100% confluence), and every second day until day 8 after induction. Primary cells were harvested at 80% confluence and at days 0, 2, 5, 8, and 13. RNA and proteins were isolated using the AllPrep DNA/RNA/Protein kit (Qiagen GmbH, Hilden, Germany) according to the manufacturer’s instructions. Gene and protein expression analyses were performed as described above.

### 2.12. Statistical Analysis

Statistical analyses were performed using Prism 6.0 software (GraphPad Software, San Diego, CA, USA). Data are given as the means ± SD or SEM. Data were analyzed using a two-tailed unpaired Student’s *t*-test or one-way ANOVA. *p*-values < 0.05 were considered statistically significant.

## 3. Results

### 3.1. Generation of ATHoxc9^-/-^ Mice

AT*Hoxc9*^-/-^ mice were generated by crossing mice carrying the loxP-flanked *Hoxc9* allele with transgenic mice expressing Cre recombinase under the control of the adipocyte-specific *Fabp4* promoter. The knockout strategy is shown in Figure 1A. Mice were genotyped by PCR of genomic DNA followed by agarose gel electrophoresis. According to the PCR product, mice were classified as wild-type (*Hoxc9*^wt/wt^, 205 bp), heterozygous (*Hoxc9*^lox/wt^, 205 bp and 324 bp), or homozygous (*Hoxc9*^lox/lox^, 324 bp) for the loxP-flanked *Hoxc9* allele (Figure 1B). Furthermore, mice were genotyped for the presence of Cre recombinase (Figure 1B). *Hoxc9*^lox/lox^ mice were used as controls (Ctrl), whereas mice presenting both Cre recombinase and homozygous loxP-flanked *Hoxc9* were considered knockouts (AT*Hoxc9*^-/-^, KO, Figure 1B). 

### 3.2. Limitations of Fabp4-Cre-Mediated Hoxc9 Targeting

We systematically assessed *Hoxc9* knockdown efficiency on DNA, RNA, and protein levels (Figure 2). Genotyping of AT from Ctrl and AT*Hoxc9^-/-^* mice could be clearly distinguished from *Hoxc9^lox/lox^* and *Fabp4-Cre*^-^ controls (Figure 2A). However, qRT-PCR analyses of *Hoxc9* mRNA expression in BAT, ingWAT, and eWAT showed highly heterogeneous expression patterns in both AT*Hoxc9*^-/-^ and control mice (Figure 2B). Finally, in Western Blot analyses, the clearly distinct Hoxc9 genotype was not associated with significantly lower Hoxc9 protein levels in WAT of AT*Hoxc9*^-/-^ compared to control mice (Figure 2C). Assuming that at the end of study, we could confirm Hoxc9 ablation in a sufficient number of AT*Hoxc9*^-/-^ mice, we systematically characterized the phenotype of AT*Hoxc9*^-/-^ compared to control mice.

### 3.3. Male ATHoxc9^-/-^ Mice are Partially Protected Against Diet-Induced Obesity

Under chow diet conditions, AT*Hoxc9*^-/-^ and control mice of both genders exhibited normal growth until the age of 30 weeks (Figure 3A and Appendix A). Male AT*Hoxc9^-/-^* mice fed with HFD for 20 weeks gained significantly less weight compared to control mice (Figure 3A). However, relative tissue weights and body fat mass were not significantly different between AT*Hoxc9^-/-^* and control mice after HFD (Figure 3B,C). Lean mass, as well as body length were not different between AT*Hoxc9^-/-^* and control mice or as a function of different diets (Figure 3D–E). In contrast, female AT*Hoxc9*^-/-^ mice were not different from controls regarding body weight or fat mass (Appendix A). Daily food intake was indistinguishable between AT*Hoxc9*^-/-^ and control mice in both sexes (Figure 3F and Appendix A). 

### 3.4. Consequences of ATHoxc9^-/-^ on AT Morphology and Inflammation

Histological analyses of both ingWAT and eWAT revealed that AT*Hoxc9*^-/-^ mice had significantly smaller adipocytes under chow diet (CD) conditions (Figure 4A–C). Under HFD, adipocytes from eWAT of AT*Hoxc9*^-/-^ mice became significantly larger compared to those of control mice, whereas no adipocyte size difference was detectable between the genotypes in ingWAT (Figure 4E–H). Analyses of adipocyte size distribution only revealed a genotype difference in ingWAT and eWAT uponCD (Figure 4B–D,F–H). There were no differences in AT macrophage or CD4^+^ and CD8^+^ lymphocyte numbers, nor the ratio of M1 to M2 macrophage subpopulations in eWAT between AT*Hoxc9*^-/-^ and control mice (for males: Figure 4I–K and for females: Appendix A). In both sexes, the total number of leukocytes (CD45 positive cells) was not different between AT*Hoxc9*^-/-^ and control mice. Moreover, we did not find histological evidence for differences in AT immune cell infiltration between AT*Hoxc9*^-/-^ and control mice after HFD by conventional H&E staining. Based on that observation, we did not perform additional AT immunohistochemistry analyses. 

### 3.5. Male ATHoxc9^-/-^ Have Improved Glucose Tolerance Under High-Fat Diet

Under chow-fed conditions, the parameters of energy expenditure were not different between AT*Hoxc9*^-/-^ and control mice (Figure 5A). However, the partial protection against HFD-induced obesity in male AT*Hoxc9*^-/-^ mice could be caused by a trend towards higher energy expenditure during the light period (Figure 5B,C). In female AT*Hoxc9*^-/-^ mice, energy expenditure and activity levels were indistinguishable from control littermates (Supplementary Appendix A). In male AT*Hoxc9*^-/-^ mice, spontaneous activity was higher under chow, whereas under HFD conditions, running distance was significantly lower compared to the control (Figure 5D). Rectal body temperature measurements did not reveal genotype differences between male mice under the same dietary conditions (Ctrl vs AT*Hoxc9*^-/-^: CD 35.40 ± 0.08 °C vs. 34.50 ± 1.62 °C, and HFD 36.88 ± 1.00 °C vs. 36.66 ± 0.67 °C). Lower body weight in male AT*Hoxc9*^-/-^ mice upon HFD was associated with improved parameters of glucose metabolism including better glucose tolerance compared to controls at 25 weeks of age (Figure 5E and Table 1). However, HbA1c was not different between the groups (Table 1). Although insulin tolerance was not different between AT*Hoxc9*^-/-^ and control mice, euglycemic-hyperinsulinemic clamp studies revealed a trend for improved insulin sensitivity in male AT*Hoxc9*^-/-^ mice under chow diet (Figure 5F). We did not find significant AT*Hoxc9* genotype-related differences in circulating insulin, leptin, leptin-to-body weight ratio, adiponectin, nor serum lipid concentrations (Table 1). 

These results were consistent with AT *Adipoq* and *Lep* mRNA levels of male HFD mice (Table 2). *Plin1* levels were higher in AT*Hoxc9*^-/-^ male mice under both diets. Under CD, only five out of 91 OLINK panel analyzed circulating proteins were different between the genotypes, whereas under HFD conditions, 18 parameters discriminated AT*Hoxc9*^-/-^ from control mice (Table 1).

### 3.6. Fabp4 is Expressed Later than Hoxc9 During Adipogenesis

Against our expectation from previously characterized *Fabp4*-Cre-mediated models [17,18], we did not find a sufficient number of AT*Hoxc9*^-/-^ mice with a biologically relevant ablation of *Hoxc9* in adipose tissue. We therefore systematically analyzed expression patterns of *Hoxc9* and *Fabp4* during adipogenesis both in different adipogenesis model systems in vitro and in ingWAT from newborn wild-type mice. Consistently, we found in immortalized adipocytes, 3T3-L1 cells, primary preadipocytes from wild-type mice and preadipocytes differentiated from newborn mice a maximum of *Hoxc9* mRNA expression either at the 80% confluent stage or during initiation of differentiation at day 0 (Figure 6A–C). In contrast, induction of *Fabp4* mRNA expression during adipogenesis only started between day 1 and day 2 (Figure 6A–C). In mature adipocytes, *Hoxc9* mRNA was only detectable at low expression levels, reaching a peak expression in ingWAT at days 5 and 10 after birth (Figure 6D). In contrast, *Fabp4* was lowly expressed from day 0 to day 10 and rose after day 15, when *Hoxc9* mRNA was already dropped (Figure 6D). Differences between Hoxc9 and Fabp4 AT protein abundance during adipogenesis further implicated that *Fabp4*-mediated targeting strategies were not suitable to target genes expressed early during adipocyte development (Figure 6E). Whereas Fabp4 protein abundancy continuously increased with adipocyte maturation, Hoxc9 protein expression peaked between days 0 and 2 and from day 15 to day 20 with a nadir at day 5 and day 10 (Figure 6E).

## 4. Discussion

Developmental genes including *Hoxc9* play an important role in the regulation of AT distribution and function [6,7,19]. Moreover, the anteroposterior expression pattern of *Hoxc9* in mice indicates a relationship between anatomic localization, AT identity, and function [20]. Furthermore, it has been demonstrated that *Hoxc9* is specifically expressed in white, but not in brown AT [6,21,22]. We therefore tested the hypothesis that AT-specific ablation of *Hoxc9* in male and female AT*Hoxc9^-/-^* mice affects AT morphology, distribution, and function. 

In general, *Hoxc9* expression in AT was low and highly variable in both sexes, across different fat depots, but also in mice with the clearly distinguishable AT*Hoxc9^-/-^* genotype. Despite the heterogeneous efficacy of the *Fabp4*-Cre-mediated AT*Hoxc9* knockout, we continued our experiments comparing AT*Hoxc9*^-/-^ to the control genotype. This decision was based on the assumption of obtaining a sufficient number of mice with the AT*Hoxc9*^-/-^ genotype and an effective ablation of AT*Hoxc9* to be included in our analyses. However, at the end of the studies, we were not able to select mice, which allowed discriminating the *Hoxc9* genotype at the AT*Hoxc9* expression level. 

Despite these shortcomings of the model, we found that male AT*Hoxc9^-/-^* mice were partially protected against weight gain in response to HFD. Interestingly, reduced *Hoxc9* expression was associated with a higher degree of adipocyte hypertrophy in eWAT after HFD. This observation supported human data describing negative correlations between *Hoxc9* expression and adipocyte size even after adjusting for body fat mass [7]. Cells of the stromal vascular fraction express considerable amounts of *Hoxc9* [6,7,23]. It is therefore important to note that the number of macrophages in AT was not different between AT from AT*Hoxc9^-/-^* and control mice.

Partial protection against HFD-induced weight gain in male AT*Hoxc9^-/-^* mice was associated with improved glucose tolerance compared to controls. This result was in contrast to our findings in humans suggesting that lower AT *Hoxc9* expression may be related to a phenotype with impaired glucose metabolism and insulin sensitivity [7]. 

The factors causing lower body weight in male AT*Hoxc9^-/-^* mice were further explored following recently published guidance for the analysis of mouse energy metabolism [24]. Because food intake was not significantly different between AT*Hoxc9^-/-^* and control mice, we investigated energy expenditure in metabolic chambers. Indeed, AT*Hoxc9^-/-^* mice displayed higher energy expenditure during the light phase, potentially underlying lower body weight gain upon HFD. In contrast, lower running activity and indistinguishable basal body temperature and brown AT mass did not explain the leaner phenotype of AT*Hoxc9^-/-^* after HFD.

Leptin secretion plays a role in the regulation of whole body energy metabolism and may contribute to the previously reported mouse strain differences in the response to HFD [25]. Moreover, leptin can stimulate locomotor activity [26]. However, circulating leptin levels were not different between AT*Hoxc9^-/-^* and control mice. Taken together, our data suggest a role of *Hoxc9* in the development of obesity and the determination of fat depot-specific signatures, which may affect whole body glucose metabolism, insulin sensitivity, and energy expenditure.

Independent of the diet, female AT*Hoxc9^-/-^* mice were not significantly different from controls with regard to body weight dynamics, parameters of AT function, morphology, and glucose metabolism. Sex-related factors such as circulating sex hormones and sex-hormone receptor expression [20] or differences in adipocytes metabolism and function [27] may explain the observed subtle sex differences in the phenotype of AT*Hoxc9^-/-^* mice. 

Since the first mammalian Cre recombinases were used for gene targeting in 1992, Cre-loxP strategies have been commonly used to analyze gene functions [28,29,30,31,32]. There are several Cre recombinase models for the study of AT, which vary in efficiency and specificity for adipocytes [33]. Based on previous experience and the wide use of the model [9], we chose a *Fabp4*-Cre recombinase-mediated approach to target *Hoxc9.* We found that *Hoxc9* AT expression was highly variable and not strictly correlated with the AT*Hoxc9^-/-^* genotype. The inefficiency of the *Fabp4*-Cre model has been reported for other target genes previously [31]. Importantly, analyses of animals selected for more efficient AT*Hoxc9* ablation did not reveal different phenotypes compared to the analyses by genotype.

The detected low *Hoxc9* AT expression levels were consistent with previously reported AT *Hoxc9* expression levels [32,34]. Additionally, *Fabp4*-Cre is expressed relatively late during adipogenesis [35,36]. Here, we analyzed endogenous expression patterns of both *Hoxc9* and *Fabp4* in an ex vivo and three independent in vitro model systems. All tested adipocyte models revealed that *Hoxc9* was already expressed at early stages of adipogenesis, whereas *Fabp4* expression only increased later during adipogenesis. This finding supported a recent argument that *Fabp4*-Cre is not suitable to investigate genes that exert their main effects during the early phase or even as initiation factors of adipogenesis [37]. Searching for an alternative Cre recombinase to target early expressed genes in (pre-) adipocytes, we examined adiponectin (*Adipoq*), as well as resistin (*Retn*) gene expression in our in vitro models (Appendix A). However, both alternative potential Cre promoter lines appeared to be equally unsuitable with regard to the expression profiles in relation to *Hoxc9*. Therefore, we would not expect a different outcome using the highly AT-specific *Adipoq*-Cre [38] or the less commonly used *Retn*-Cre [39]. Berry and Rodeheffer could demonstrate that white adipocyte precursor cells (APCs) were characterized by platelet-derived growth-factor receptor, alpha polypeptide (*Pdgfra*) expression [40]. These data suggested *Pdgfra*-Cre recombinase as an interesting target to delete *Hoxc9* at early adipogenesis stages. Despite the fact that this model was not available to our group in the project planning phase, there are additional scientific caveats related to the use of this model. *Pdgfra* is expressed in several cells other than adipocytes, including tissues of neuroectodermal or mesenchymal origin [37], hepatocytes, and skeletal muscle cells [33]. 

Despite the shortcomings of the *Fabp4*-Cre-mediated *Hoxc9* targeting, we found a genotype-phenotype association in male AT*Hoxc9^-/-^* mice after HFD-induced obesity. We therefore proposed that a moderate reduction in *Hoxc9* gene expression may be sufficient to cause a lower weight gain in response to HFD. On the other hand, we could not exclude that the observed phenotype may be caused by independent and unknown mediators. In this context, it has been described that genetically identical C57BL/6J mice respond with a high degree of body weight variation to HFD [41]. Specific genotype-environment interactions [25] or variability in the intestinal uptake [42] may underlie heterogeneous response to diet challenges. In other studies, individual body weight differences in HFD response of C57BL/6J mice ranged from 27.2 to 52.7 g [43]. In addition, we could not exclude that differences in expression and activity of *Fabp4*-Cre recombinase may account for the heterogeneous efficacy in AT Hoxc9 ablation. It is noteworthy, that we did not formally prove this hypothesis, because *Fabp4*-Cre-expressing mice were not systematically studied as controls. However, decreased responsiveness for diet-induced obesity has been a repeated finding in *Fabp4*-Cre mice [44,45,46]. 

## 5. Conclusions

In conclusion, we found that crossing *Hoxc9* floxed mice with *Fabp4*-Cre mice did not produce a biologically relevant ablation of *Hoxc9* in adipose tissue and could therefore not be used to analyze the impact of *Hoxc9* on body weight regulation or in vivo adipose tissue function. Nevertheless, our data did not exclude a role of *Hoxc9* in the development of obesity, AT distribution, and adipocyte function. The limitations of the *Fabp4*-Cre targeting strategy need to be considered particularly in the generation of models to investigate the function of developmental genes in AT.

## Figures and Tables

**Figure 1 biomedicines-08-00184-f001:**
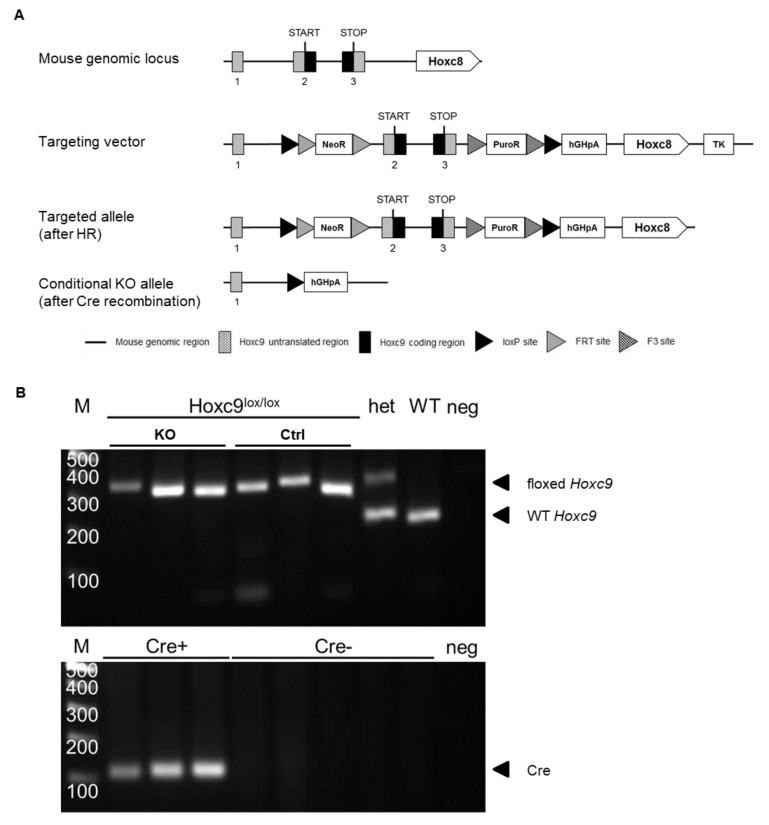
Targeting strategy for the generation of AT*Hoxc9*^-/-^ mice. (**A**) Schematic representation of the genomic *Hoxc9* locus with the neighboring *Hoxc8* locus, the targeting vector, and the loxP-flanked *Hoxc9* allele after homologous recombination (HR) before and after crossing with transgenic mice expressing Cre recombinase under the control of the *Fabp4* promotor. The targeting vector consists of a 5.3 kb loxP-flanked region containing *Hoxc9* exons 2 and 3, a thymidine kinase cassette (TK) as a negative selection marker, and two positive selection markers: a neomycin resistance cassette (NeoR) flanked by FRT sites and a puromycin resistance cassette (PuroR) flanked by F3 sites. The knockout allele (KO) is characterized by loss of *Hoxc9′*s coding region located in exons 2 and 3. (**B**) Agarose gel electrophoresis after PCR of genomic DNA from homozygous floxed *Hoxc9* (*Hoxc9*^lox/lox^, predicted PCR product of 324 bp), wild-type (WT, *Hoxc9*^wt/wt^, predicted PCR product of 205 bp), heterozygous *Hoxc9* (*Hoxc9*^lox/wt^, predicted PCR products of 205 and 324 bp), and *Fabp4*-Cre^+^ mice (Cre^+^, predicted product of 100 bp). Negative control lane without genomic DNA (neg). Arrows indicate PCR products.

**Figure 2 biomedicines-08-00184-f002:**
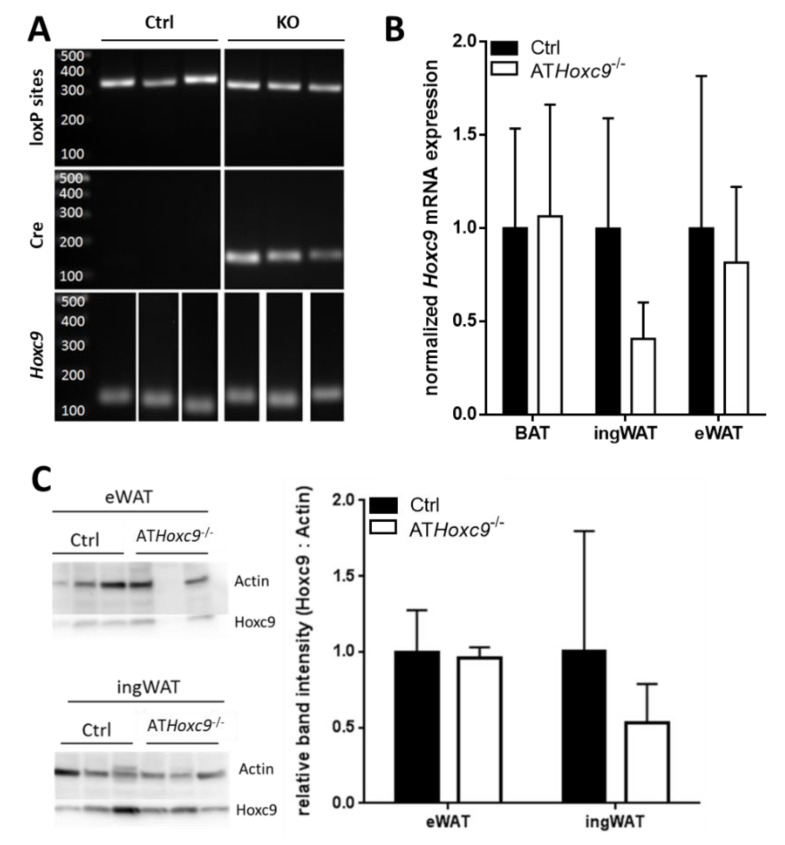
Detection of *Hoxc9* in adipose tissue. (**A**) Agarose gel electrophoresis of PCRs performed with genomic DNA isolated from inguinal white adipose tissue (ingWAT) of control (*Hoxc9*^lox/lox^ and *Fabp4*-Cre^-^) and AT*Hoxc9*^-/-^ (*Hoxc9*^lox/lox^ and *Fabp4*-Cre^+^) mice. Both control and AT*Hoxc9*^-/-^ mice showed a heterozygous *Hoxc9*^lox/lox^ product of 324 bp representing floxed *Hoxc9* exons 2 and 3 (loxP sites). Products of ~100 bp revealing *Fabp4*-Cre recombinase were just present in AT*Hoxc9*^-/-^ animals (Cre). Finally, a product of 115 bp illustrating a DNA fragment in between the *Hoxc9* intron 1 exon 2 junction was detectable in both control and AT*Hoxc9*^-/-^ mice (Hoxc9). (**B**) *Hoxc9* gene expression patterns in murine AT depots were measured using LightCycler 480 and SybrGreen I assay relative to *36B4* expression and normalized to *Hoxc9*^lox/lox^ animals. Expression values were calculated according to Pfaffl et al. (2001) [12]. *n* = 8 per tissue and genotype. Both sexes and diets included. (**C**) Protein levels in epididymal white AT (eWAT) and ingWAT were determined using western blot followed by densitometric analysis using GeneTools 4.3.8 software (Syngene, Cambridge, UK). Hoxc9 band intensities are presented relative to actin levels and normalized to *Hoxc9*^lox/lox^ mice.

**Figure 3 biomedicines-08-00184-f003:**
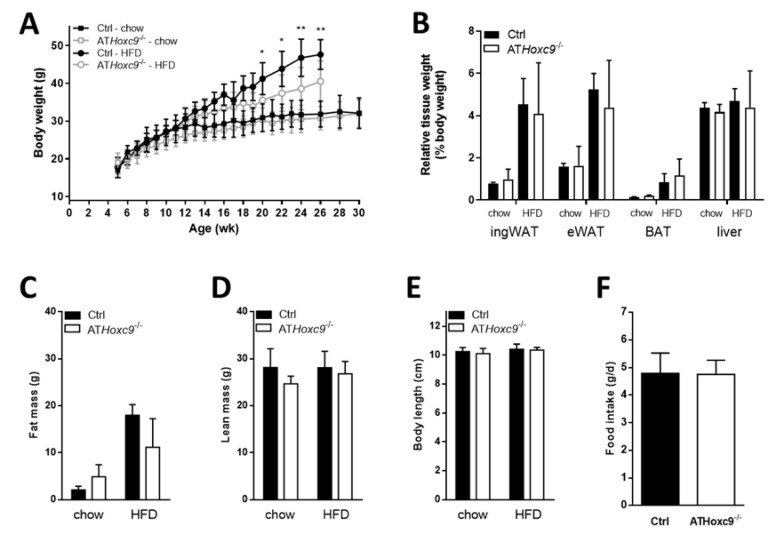
Phenotyping of male AT*Hoxc9* deficient mice. (**A**–**F**) Body weight gain, relative tissue weights, fat and lean mass, as well as body length and food intake do not differ between AT*Hoxc9*^-/-^ and control male mice, except body weight differences under high-fat diet (HFD). * *p* < 0.05, ** *p* < 0.01. BAT, brown AT.

**Figure 4 biomedicines-08-00184-f004:**
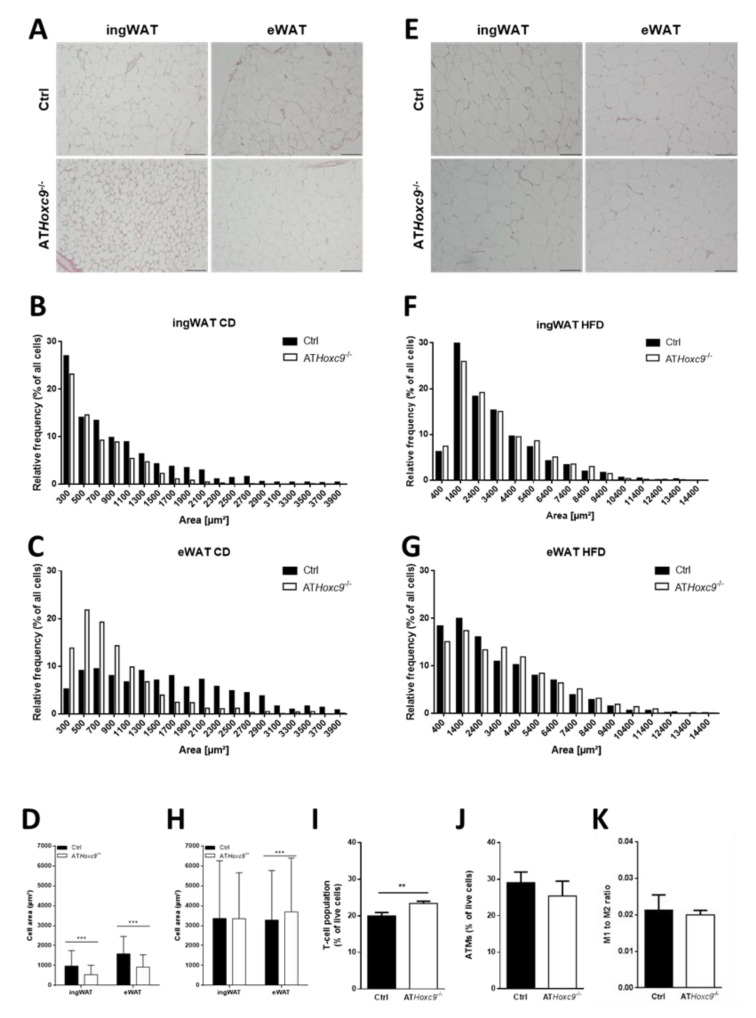
Consequences of *Hoxc9* targeting on adipose tissue morphology. (**A**) H&E staining of ingWAT and eWAT under chow diet (CD) Ctrl vs. AT*Hoxc9*^-/-^. Ten times magnification, bar size = 100 µm. (**B**,**C**) Adipocyte area measurements represented as fractions of total amount of counted cells. (**D**) Cell area measurements of ingWAT and eWAT from male mice under CD calculated with BZ-X800 Analyzer Software. n(ingWAT, Ctrl) = 1408, n(ingWAT, AT*Hoxc9*^-/-^) = 1676, n(eWAT, Ctrl) = 752, n(eWAT, AT*Hoxc9*^-/-^) = 1192. (**E**) H&E staining of ingWAT and eWAT under HFD Ctrl vs. AT*Hoxc9*^-/-^. Ten times magnification, bar size = 100 µm. (**F**,**G**) Adipocyte area measurements represented as fractions of total amount of counted cells. (**H**) Cell area measurements of ingWAT and eWAT from male mice under HFD calculated with BZ-X800 Analyzer Software. n(ingWAT, Ctrl) = 1137, n(ingWAT, AT*Hoxc9*^-/-^) = 1017, n(eWAT, Ctrl) = 1394, n(eWAT, AT*Hoxc9*^-/-^) = 1337. (**I**–**K**) Analysis of the immune phenotype shows no differences with respect to adipose tissue macrophage (ATM) populations and the M1 to M2 macrophage ratio in lean male mice. Viable CD45^+^ (CD, Cluster of differentiation) and F4/80^+^ cells were defined as ATMs. Subpopulations could be further distinguished into M1 (CD11c^+^; CD206^−^) and M2 (CD11c^−^; CD206^+^). Viable CD45^+^ and CD3^+^ cells were defined as T lymphocytes, further divided into T helper cells (T_H_; CD4^+^; CD8^−^) and cytotoxic T cells (T_C_; CD4^−^; CD8^+^). Gating strategies are shown in Appendix A. *n* = 6–10 mice each group. Data represent the mean ± SEM. ** *p* < 0.01, *** *p* < 0.001.

**Figure 5 biomedicines-08-00184-f005:**
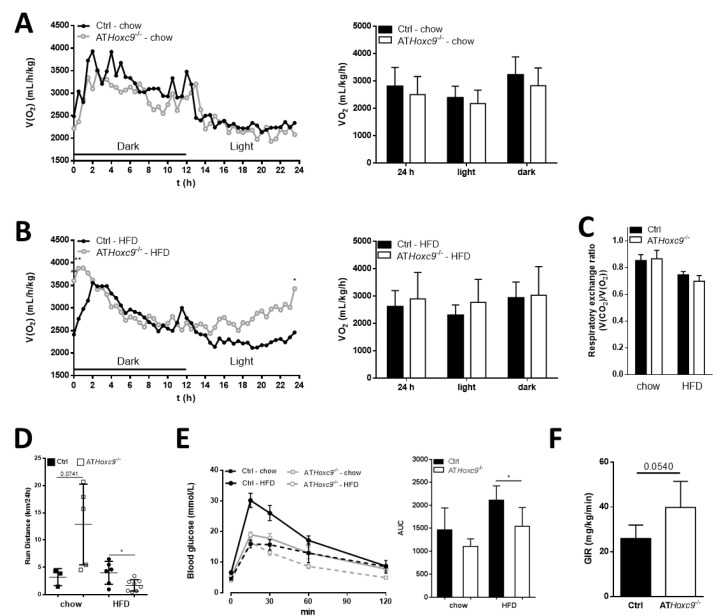
Metabolic parameters of AT*Hoxc9* deficient mice. (**A**–**C**) Oxygen consumption and respiratory exchange rate (RER, **C**) were not altered by the presence or absence of *Hoxc9* in AT in chow (**A**) or in HFD fed mice (**B**). (**D**) Run distance measurements revealed contrary results with respect to diet. Whereas chow-fed AT*Hoxc9*^-/-^ mice tended to run more than littermate controls, they behaved oppositely under HFD. (**E**) Male AT*Hoxc9*^-/-^ mice showed better glucose tolerance under HFD compared to control animals during intraperitoneal glucose tolerance tests (GTT) after 25 weeks of age. (**F**) Hyperinsulinemic-euglycemic clamps were performed in chow diet animals at 23–25 weeks of age to determine insulin sensitivity represented by the glucose infusion rate (GIR,). *n* (**A**–**C**) = 6–8, *n* (**D** and **E**) = 4–10, *n* (**F**) = 3–10 mice each group. Data represent the mean ± SEM. * Significantly different between Ctrl and AT*Hoxc9*^-/-^ at *p* < 0.05.

**Figure 6 biomedicines-08-00184-f006:**
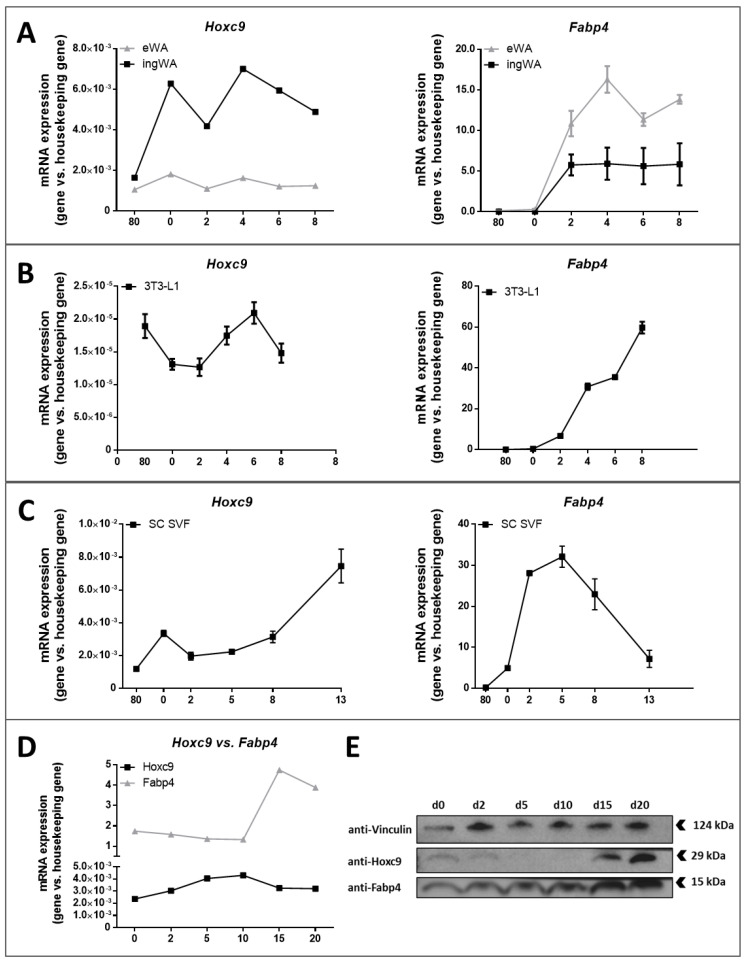
Gene and protein expression patterns of *Hoxc9* and *Fabp4* in different in vitro and in vivo models. All gene expression experiments were performed using LightCycler 480 and SybrGreen I assay. Mapped time points were at 80% pre-confluence (80), 100% confluence, and initiation of differentiation (0) and days post-confluence (2–8 or 13, respectively). Expression values were relative to housekeeping gene expression levels. (**A**) Immortalized inguinal and epididymal adipocytes. Gene expression levels were related to *36B4* and *Actb* expression patterns. (**B**) 3T3-L1 cell line. Gene expression levels were related to *36B4* and *Actb* expression patterns. (**C**) SVF cells isolated from ingWAT of C57BL/6NTac mice. Gene expression levels were related to *36B4*, *Ppia*, and *Himbs* expression patterns. *n* = 2. (**D**) Gene expression levels of *Hoxc9* and *Fabp4* in ingWAT. Expression levels were related to *36b4* and *Actb* expression patterns. *n* = 2 animals per time point. Data represented as the mean ± SEM. (**E**) Representative Western blot of newborn male mice to determine protein levels during in vivo adipogenesis. Arrows represent the expected product size.

**Table 1 biomedicines-08-00184-t001:** Serum concentrations of the parameters of lipid metabolism and glucose homeostasis. Measured in male mice at an age of 30 (CD) or 26 weeks (HFD).

	Chow Diet (CD)			High Fat Diet (HFD)		
	CD Ctrl	ATHoxc9^-/-^	*n*	HFD Ctrl	ATHoxc9^-/-^	*n*
**Serum Lipids**						
TGs (mmol/L)	0.79 ± 0.01	0.78 ± 0.17	3 vs. 10	1.32 ± 0.37	1.34 ± 0.35	8 vs. 9
Cholesterol (mmol/L)	2.23 ± 0.29	2.03 ± 0.28	3 vs. 10	3.76 ± 1.05	4.24 ± 0.68	8 vs. 9
HDL cholesterol (mmol/L)	2.23 ± 0.26	2.06 ± 0.38	3 vs. 10	3.26 ± 1.29	3.34 ± 0.43	8 vs. 9
LDL cholesterol (mmol/L)	0.26 ± 0.05	0.23 ± 0.08	3 vs. 10	0.65 ± 0.36	0.82 ± 0.23	8 vs. 9
FFA (mmol/L)	0.77 ± 0.38	0.98 ± 0.21	3 vs. 10	1.10 ± 0.10	1.33 ± 0.24	8 vs. 9
Ahr				3.62 ± 0.54	**4.70 ± 0.14****	3 vs. 4
Axin1				2.94 ± 0.36	**4.24 ± 0.20*****	3 vs. 4
Ca13				4.50 ± 0.25	**5.46 ± 0.16***	3 vs. 4
Ccl5				4.21 ± 1.13	**2.64 ± 0.28*****	3 vs. 4
Ddah1				2.66 ± 0.68	**4.40 ± 0.24*****	3 vs. 4
Fli1				2.10 ± 0.38	**3.81 ± 0.31*****	3 vs. 4
Flrt2				6.95 ± 0.37	**7.91 ± 0.17***	3 vs. 4
Ghrl	5.05 ± 0.97	**3.46 ± 0.77****	3 vs. 4			
Il1a	6.10 ± 0.35	**7.52 ± 0.68***	3 vs. 4			
Il5	2.16 ± 0.77	**0.70 ± 0.84***	3 vs. 4			
Il17a				5.10 ± 0.46	**3.83 ± 0.58*****	3 vs. 4
Il17f				5.94 ± 0.79	**4.51 ± 0.75*****	3 vs. 4
Itgb1bp2				3.27 ± 0.72	**4.31 ± 0.20****	3 vs. 4
Pak4				1.08 ± 0.28	**2.13 ± 0.21****	3 vs. 4
Parp1	4.24 ± 0.59	**5.61 ± 0.94***	3 vs. 4			
Pla2g4a				8.05 ± 0.13	**8.91 ± 0.16***	3 vs. 4
Plin1	3.59 ± 1.06	**5.25 ± 1.34****	3 vs. 4	6.77 ± 0.37	**7.67 ± 0.22***	3 vs. 4
Riox2				5.87 ± 0.24	**6.99 ± 0.15*****	3 vs. 4
Snap29				6.70 ± 0.28	**7.57 ± 0.20***	3 vs. 4
Tgfa				7.25 ± 0.15	**8.36 ± 0.23*****	3 vs. 4
Tnni3				13.68 ± 0.34	**12.73 ± 0.86***	3 vs. 4
Yes1				3.68 ± 0.42	**5.08 ± 0.30*****	3 vs. 4
**Glucose Homeostasis**						
Insulin (ng/mL)	0.25 ± 0.07	0.23 ± 0.10	2 vs. 6	2.89 ± 2.17	1.23 ± 0.76	6 vs. 6
Adiponectin (µg/mL)	72.12 ± 2.34	71.86 ± 20.82	3 vs. 10	62.74 ± 7.70	60.86 ± 7.72	4 vs. 4
Leptin (ng/mL)	1.15 ± 0.34	1.48 ± 0.89	3 vs. 10	45.71 ± 14.71	29.50 ± 21.53	6 vs. 6
Leptin/body weight (ng/mL/g)	0.039 ± 0.011	0.053 ± 0.029	3 vs. 10	1.08 ± 0.38	0.76 ± 0.52	6 vs. 6
Fasting glucose (mmol/L)	4.17 ± 0.62	**3.44 ± 0.40***	3 vs. 10	6.44 ± 1.06	5.62 ± 1.88	5 vs. 6
HbA1c (%)	4.67 ± 0.24	4.44 ± 0.39	2 vs. 10	4.35 ± 0.10	4.48 ± 0.18	7 vs. 7

All values were obtained after a 16 h overnight fasting period. Significantly different values highlighted in bold. * Significantly different between Ctrl and KO animals of the same diet at * *p* < 0.05, ** *p* < 0.01, *** *p* < 0.001. CD = chow diet, HFD = high-fat diet, Ctrl = *Hoxc9*^lox/lox^, KO = AT*Hoxc9*^-/-^, TGs = triglycerides, HDL = high density lipoprotein, LDL = low density lipoprotein, FFA = free fatty acids, HbA1c = glycated hemoglobin, AU = arbitrary units.

**Table 2 biomedicines-08-00184-t002:** Gene expression measurements in ingWAT and eWAT of HFD male mice. Gene expression was normalized for *36b4* gene expression as the housekeeping gene.

	Ctrl	KO	*n*
**ingWAT**			
*Adipoq*	26.19 ± 9.16	24.88 ± 12.10	5 vs 6
*Dlk1*	0.047 ± 0.009	0.043 ± 0.017	5 vs 6
*Fabp4*	119.8 ± 35.53	**85.14 ± 27.56 ****	5 vs 6
*Ki67*	0.005 ± 0.002	0.004 ± 0.003	5 vs 6
*Lep*	3.87 ± 1.93	1.56 ± 1.76	5 vs 6
*Plin1*	2.49 ± 1.12	2.84 ± 1.32	5 vs 6
**eWAT**			
*Adipoq*	25.41 ± 10.57	29.30 ± 9.78	5 vs 6
*Dlk1*	0.029 ± 0.022	0.027 ± 0.007	5 vs 6
*Fabp4*	89.47 ± 18.99	97.19 ± 20.20	5 vs 6
*Ki67*	0.009 ± 0.004	0.009 ± 0.003	5 vs 6
*Lep*	4.98 ± 1.52	2.58 ± 1.72	5 vs 6
*Plin1*	1.73 ± 0.41	2.27 ± 0.76	5 vs 6

Significantly different values highlighted in bold. ** Significantly different between Ctrl and KO animals of the same diet at *p* < 0.01. ingWAT = inguinal white adipose tissue, eWAT = epigonadal adipose tissue, HFD = high-fat diet, Ctrl = *Hoxc9*^lox/lox^, KO = AT*Hoxc9*^-/-^.

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
