# Peer review of "The Fabp4-Cre-Model is Insufficient to Study Hoxc9 Function in Adipose Tissue"

_biomedicines, 2020, doi:10.3390/biomedicines8070184_

Round 1

Reviewer 1 Report

Brief Summary:

Authors aimed to study the role of Hox9 in adipose tissue development in Fabp4-C mediated Hox9 knockout. Although the topic is very interesting the manuscript is not well written and some data seem confusing and some alterations are required as follows:

General comments:

Line 22-23: I cannot understand the following sentence “However, this phenotype could not be observed under CD conditions”. Please revise it.

Results are not clear in abstract section. The context of the study is also missing.

Keywords: I suggest to change the keywords as they should not be in the title.

In your studies you analyse male and female, but in abstract these findings are not mentioned.

In the table 1, you used the term ctrl twice. In fact, one of these should be changed to HFD.

Why did you used different times at an age of 30 (CD) or 26 weeks (HFD)? The explanation is needed.

Leptin values are not significant between HFD and Hox9 knockout and the same for leptin/BW, any explanation?

Reviewer 2 Report

The current manuscript by Dommel et al. is interesting and have investigated the role of adipose-specific Hoxc9 in obesity and associated metabolic pathways. Its a continuation of previous human study about Hoxc9 and 10 (PMID: 2664790). Although the manuscript is interesting and technically sound, the current manuscript have major flaws mainly in experimental set up and presentation of the obtained results. They are as follows:

  1. Title is confusing and too descriptive without a clear message. Authors need to decide on either ATHoxc9 function or inefficiency of Fabp4 cre.
  2. The whole manuscript not easy to read. Throughout the manuscript results are overstated. Eg: Line 394; the fat mass are not reduced significantly in KO, there is merely a downward pattern like most cases.
  3. Line 405-407, apart from iBAT weight and rectal temp; the authors have not considered about the higher lean mass (Fig 2D) in -HFD-KO which can explain higher energy expenditure of HFD-KO. 
  4. The authors should include representative gating strategy for immune cells flow cytometry. Currently its not clear at all, Eg: Fig 3I,J-% of SVC-what are the markers? Are they gated from live cells? Fig 3I-Are these CD45+ cells? And lastly immunophenotyping of HFD-fed mice would have answered more questions about inflammatory status and its role in glucose-dysmetabolism. 
  5. Its kind of compulsory to check for the knock-out at both mRNA and protein levels at the start of the project/experiment, rather than at the end. However, in Fig 5B the authors should include multiple WATs and BATs to show the effectiveness of Fabp4 Cre.
  6. What can be the reason behind inefficiency of Fabp4 Cre in females only? Fig 6E, Are these pups are male or female (d10,15,20)? From the immunolbot its clear that even in d0 there is a significant Fabp4 expression. By the way the authors should also add loading control immunoblot and run ATHoxc9 KO side by side.
  7. Discussion is too long and repetitive with some part of results. 

Round 2

Reviewer 2 Report

The current version is highly improved but still have some errors as following

  1. Fig S1 legend: The authors should use epididymal WAT or epigonadal. And what are these eWA and ingWA?
  2. Flow gating strategy is not clear, especially when comparing with flow figure in Fig 4. Depending on the gating strategy Fig 4I can't be lymphocytes. Is these T-cells/T lymphocytes? The authors should also include total immune cells CD45, and different T cell subpopulations.
  3. 'we did not see differences in AT immune cell infiltration comparing AT histology from ATHoxc9-/- and control mice' is these done by IHC. If yes, adding those figures in the revised version would definitely help.
